# Fingerprinting of Volatile Organic Compounds for the Geographical Discrimination of Rice Samples from Northeast China

**DOI:** 10.3390/foods11121695

**Published:** 2022-06-09

**Authors:** Sailimuhan Asimi, Xin Ren, Min Zhang, Sixuan Li, Lina Guan, Zhenhua Wang, Shan Liang, Ziyuan Wang

**Affiliations:** 1Beijing Advanced Innovation Center for Food Nutrition and Human Health, Beijing Technology and Business University, Beijing 100048, China; salima2@126.com (S.A.); renxin@btbu.edu.cn (X.R.); lisixuan1929@163.com (S.L.); m15046782736@163.com (L.G.); zhwang@btbu.edu.cn (Z.W.); liangshan@btbu.edu.cn (S.L.); wangziyuan@btbu.edu.cn (Z.W.); 2Beijing Engineering and Technology Research Center of Food Additives, Beijing Technology and Business University, 11 Fucheng Road, Beijing 100048, China

**Keywords:** rice, HS-GC-MS, volatile organic compound, geographical origin, partial least squares discriminant analysis (PLS-DA), authenticity

## Abstract

Rice’s geographic origin and variety play a vital role in commercial rice trade and consumption. However, a method for rapidly discriminating the geographical origins of rice from a different region is still lacking. Therefore, the current study developed a volatile organic compound (VOC) based geographical discrimination method using headspace gas chromatography-mass spectrometry (HS-GC-MS) to discriminate rice samples from Heilongjiang, Jilin, and Liaoning provinces. The rice VOCs in Heilongjiang, Liaoning, and Jilin were analyzed by agglomerative hierarchical clustering (AHC), principal component analysis (PCA), and partial least squares discriminant analysis (PLS-DA). The results show that the optimum parameters for headspace solid phase microextraction (HS-SPME) involved the extraction of 3.0 g of rice at 80 °C within 40 min. A total of 35 VOCs were identified from 30 rice varieties from Northeast China. The PLS-DA model exhibited good discrimination (R^2^ = 0.992, Q^2^ = 0.983, and Accuracy = 1.0) for rice samples from Heilongjiang, Liaoning, and Jilin. Moreover, K-nearest neighbors showed good specificity (100%) and accuracy (100%) in identifying the origin of samples. In conclusion, the present study established VOC fingerprinting as a highly efficient approach to identifying rice’s geographical origin. Our findings highlight the ability to discriminate rice from Heilongjiang, Liaoning, and Jilin provinces rapidly.

## 1. Introduction

Rice is one of the important cereals that provide energy for human beings. Therefore, more than half of the world’s population eats rice as a staple food [1]. Northeast China is the most important commercial rice production base in China, and northeast rice is the most popular rice among consumers. Northeast rice has been cultivated in fertile black soil, and the unique climate of Northeast China provides enough light and water. It has a good taste and rich nutrition and is widely loved by people [2]. Rice food quality is often associated with geographical sources, the primary consideration for rice consumers [3]. Rice with a geographic origin label in China, such as Wuchang rice in Heilongjiang, Panjin rice in Liaoning, and Baijinxiang rice in Jilin, are considered high-quality rice and favored by consumers [4]. While there are significant differences in rice quality and composition across regions, it seems impossible to discriminate rice’s geographic origin by appearance or regular analysis [5]. Therefore, illegal delivery and misnaming for tax avoidance concern rice-consuming countries [6]. To avoid fraudulent rice trading, efficient and high-throughput identification techniques using modern analytical chemistry, which has heightened sensitivity and accuracy, are now more critical than before.

Whether accidentally or intentionally, adulteration in rice is possible from the crop harvest to the grain arrival to consumers. According to the needs of consumers and to avoid fraud, researchers have proposed different analysis methods, such as targeted and nontargeted analysis, to restrain rice false description, adulteration, or fake origin labeling [7,8]. The specific origin determines the specific quality of rice. Geographical origin indications have a crucial role in the unique characteristics of rice [9]. Therefore, it has improved the economic return of rice. For the last decade, the geographical indications of rice have been determined according to physical and chemical properties such as morphology (shape, width, and length), amylose, starch, protein content, cooking quality, and element concentration [9,10,11].

In addition to these technologies, DNA-based methods have proven reliable and can detect adulteration and identify geographical indications [12]. However, a recent study reported rice fingerprint identification using spectroscopy, but this technology cannot identify the marker fingerprint [3]. Therefore, metabolomic-based methods have attracted increased focus on solving the defects of the aforementioned method of determining metabolite fingerprinting for geographical identification. Nuclear magnetic resonance spectroscopy and mass spectrometry are the more commonly used analytical techniques in metabolomics [10]. The high chromatographic resolution, assay-specific detection, metabolite quantification, and ability to identify unknown substances of GC-MS make it a suitable tool for fingerprinting studies [13]. Rice is rich in carbohydrates, vitamins, other significant metabolites, and low molecular weight secondary metabolites. In addition, volatile organic compounds (VOCs) are composited during paddy growth and development [1]. Therefore, VOC metabolomics can be utilized to distinguish the peculiarity of rice, the identifiability of rice, and the properties of rice varieties from a specific place of origin. Therefore, it has been applied to evaluate geographical origin [4,14].

VOCs are the diagnostic aspects of rice quality that can determine consumers’ acceptance. They are also essential indicators of geographical origins. Although VOC metabolites have been studied in various food authenticity studies, such as geographic [1,4,14] and botanical origin identification [15,16,17], a method for quick identification of the geographic origin of rice from a different region is still lacking. Therefore, in the current study, a non-targeted VOC metabolomics method was applied to study the phenotype of rice from Heilongjiang, Jilin, and Liaoning provinces. Specifically, HS-SPME and GC-MS and multivariate pattern recognition analysis identified the potential variation markers of VOCs in rice. The chromatographic data were analyzed by agglomerative hierarchical clustering (HCA) and principal component analysis (PCA). Partial least squares discriminant analysis (PLS-DA) was applied to develop an estimation model to discriminate samples based on geographical origin. Finally, projection of importance (VIP) scores of variables were calculated to determine the important VOCs of rice.

## 2. Material and Methods

### 2.1. Chemicals and Reagents

HS-SPME fibers Divinylbenzene/carboxen/polydimethylsiloxane (2 cm, 50/30 µm DVB/CAR/PDMS), headspace vials (20 mL), and liner (2637501) were purchased from SUPELCO (Bellefonte, PA, USA). Mixed standard C7–C30 saturated alkanes (1000 μg/mL, hexanes) were purchased from Sigma-Aldrich (St. Louis, MO, USA) and used to calculate the retention index (RI) for each compound. Internal standard (2-methyl-3-heptanone) for GC-MS analysis and n-alkane for calculating the retention index were purchased from Beijing BioDee Biotechnology Company (Beijing, China).

### 2.2. Sample Collection

Samples were purchased from seed companies in Heilongjiang, Jilin, and Liaoning. All 30 rice samples (10 per province) were analyzed from three provinces. The samples of rice varieties in each province are shown in Appendix A. Four rice samples, Longing, Suijing, Longie, and Jinhua, were collected from Heilongjiang. Five different rice samples, including Jijing, Jiyang, Tonghe, Tongyuan, and Tongke were collected from Jilin. In addition, four different varieties of rice samples, including, Liaojing, Liaoxing, Yanjing, and Yanfeng, were collected from Liaoning. The rice grain was ridged and milled by an experimental ridging machine (THU35C, SATAKE, Suzhou, China) and an experimental rice milling machine (TM05C, SATAKE, Suzhou, China). The broken rice rate is less than 10%, and the moisture content of rice is 15.5%. Therefore, according to GB/T 1354-2018, rice meets the standard of first-class commercial japonica rice. Samples were packed in polythene bags after milling and stored at −40 °C.

### 2.3. Optimization of the HS-SPME Method

Following the method of our previous study [18,19], the DVB/CAR/PDMS HS-SPME fiber was applied to rice VOC analysis. To obtain the best conditions for the HS-SPME of the VOCs in rice samples, the effect of extraction time (20–60 min), extraction temperature (50–90 °C), and sample weight (1.0–5.0 g) were investigated. The optimized method approached the one-factor-at-a-time approach (OFAT) referred to by NPK et al. [15]. The total concentration of VOCs was calculated and identified, and the optimal conditions were found by comparing the total concentration.

### 2.4. Sample Preparation

Before sample testing, we avoid interference between the samples of the extracted fiber and treat the fiber at 270 °C for 1 h. A total of 3 g of rice sample was weighed in a 20 mL SPME vial and 1 μL content was added, 0.816 μg/mL of 2-methyl-3-heptanone, and the vial was immediately sealed with PTFE/silicon-lined nuts. After sealing, the vial was thermally balanced on an automatic heating incubator for 20 min.

### 2.5. Extraction of VOCs in Rice Sample

VOCs in rice samples were extracted according to optimal HS-SPME extraction conditions. After extraction, the HS-SPME fiber was desorbed at the GC injection port at 270 °C for 5 min. 

### 2.6. GC-MS Data Acquisition

GC-MS analysis was performed according to our previous research [20]. A Gas Chromatography (7890A, Agilent, Santa Clara, CA, USA) system coupled with a Mass Spectrometer (5975C, Agilent, Santa Clara, CA, USA) was used to analyze VOCs. GC conditions: DB-WAX capillary column (30 m × 0.25 mm, 0.25 μm); carrier gas was helium, the flow rate was 1.2 mL/min; splitless injection; inlet temperature was 250 °C; the temperature program was as follows: the initial temperature was 40 °C, maintained for 3 min, increased to 200 °C at 5 °C/min, and then increased to 230 °C at 10 °C/min, and held for 3 min. Mass spectrometry conditions: electron ionization source; electron energy 70 eV; transmission line temperature 280 °C; ion source temperature 230 °C; quadrupole temperature 150 °C; mass scan range *m*/*z* 55–500. 

### 2.7. Identification of VOCs in Rice Sample 

Qualitative analysis: Agilent Mass Hunter Software (B.08.00, Agilent, Santa Clara, CA, USA) was applied for data analysis. Firstly, the mass spectrometry of volatile aroma compounds was compared with the standard mass spectrometry in the NIST 14 mass spectrometry library to identify the compounds. The RI (Retention Index) value was compared with the standard odor description for confirmation [19]. 

Quantitative analysis: the flavor component is calculated by the area normalization method of peak area ratio, according to the internal standard substance of 2-methyl-3-heptanone, and the volatile flavor compounds of the chromatographic peak area were compared, and the volatile flavor compounds relative to the internal standard substance content were calculated according to:(1)$$C=(Ci_{\;}\times \;\frac{\mathrm{Sj}}{\mathrm{Si}})\;\times \;1000/m_{o}$$
where C is the mass concentration of volatile flavor compounds/(µg/kg); C_i_ is the mass concentration of internal standard/(µg/µL); Sj is the peak area of volatile flavor compounds. Si is the peak area of internal standard and m_o_ sample weight (g).

### 2.8. Statistical Analysis

Each sample was repeated at least three times. The significant differences (*p* < 0.05) in means were analyzed statistically by one-way analysis of variance (ANOVA) and Duncan’s multiple range tests by using IBM SPSS software of version 23 (SPSS Institute, Chicago, IL, USA).

### 2.9. Multivariate Analysis

Metaboanalyst [1] (www.metaboanalyst.ca, accessed on 4 March 2022) was used for multivariate statistical analysis, such as PCA, PLS-DA, and HCA. Among them, PCA analysis was performed to better visualize all the information in the dataset. PCA analysis visualizes differences between groups by projecting the objects of the dataset into the space of the first few principal components. PLS-DA analysis allows further identification of different VOCs leading to separation between different regions [13,21]. The established PLS-DA model was verified by leave-one-out cross-validation, and the quality of the model was evaluated based on R^2^ and Q^2^ scores [22]. In addition, the model was validated with 1000 times permutation tests [22,23]. The PLS-DA model generates Variable Importance Projection (VIP) scores. VOCs with VIP values greater than 1 were recognized as potentially differential compounds [23]. HCA was performed based on the measured characteristics to identify relatively homogeneous clusters within each sample group. This study analyzed the complete dataset using HCA defined by Ward link and Euclidean distance.

## 3. Results and Discussion

### 3.1. Optimization of the HS-SPME Method

#### 3.1.1. Sample Weight

To study the effects of different sample quantities on the total concentration of rice VOCs, we chose sample weights of 1.0, 2.0, 3.0, 4.0, and 5.0 g for extraction of rice odorant active compounds. The relationship between the sample weight and the total concentration of the VOCs is shown in Figure 1A. When the sample weight was 1.0–3.0 g, the VOCs content of rice significantly increased from 39.56% to 48.57%. In general, the amount of analyte adsorbed is directly proportional to the quality of the sample [15]. Therefore, higher extraction efficiency can be obtained by increasing the sample quality. However, when the sample weight is 3.0–5.0 g, the VOCs content of rice has no significant difference. It might be because fiber overload may adversely affect the sample’s relative response to volatile compounds [15]. The sample weight of 3.0 g was chosen for further study.

#### 3.1.2. Extraction Temperature

To obtain the best extraction temperature, five temperatures (50, 60, 70, 80, and 90 °C) were studied. By heightening extraction temperature, the above concentrations of analyses are predicted to be delivered into the headspace to improve the extraction capability [24]. As shown in Figure 1B, as the extraction temperature increased from 50 to 80 °C, the VOC content of rice significantly increased from 48.84% to 69.86%. It shows that the extraction efficiency of all analytes increases with the increase in extraction temperature. Usually, an 80 °C extraction temperature is widely used to extract VOCs from rice samples [1]. Therefore, the same extraction temperature was selected in this study.

#### 3.1.3. Extraction Time

Extraction time is essential for SPME equilibrium abilities [15]. The HS-SPME extraction time was investigated at 20, 30, 40, 50, and 60 min to find the optimum extraction time. As shown in Figure 1C, when the extraction time increased from 20 to 40 min, VOCs from 62.71% increased to 67.35%. Our previous studies showed that 40 min extraction time best separates VOCs from rice [19]. Thus, we chose the same extraction time as Wang et al. [19] (40 min) in this study.

### 3.2. GC-MS Analysis

The VOCs in rice samples were extracted using the optimum extraction parameters and analyzed by GC-MS. Overall, 30 samples, including 10 from Heilongjiang, 10 from Jilin, and 10 from Liaoning, were analyzed using HS-SPME-GC–MS (Figure 2A). The VOCs are expressed as the chromatographic peaks in the total ion chromatogram (TIC) (Appendix A). A total of 33 VOCs were identified based on their respective mass spectra and RI. The detected VOCs are listed in Appendix A. Heilongjiang and Jilin have 32 identical VOCs, Jilin and Liaoning samples have 30 identical VOCs, Heilongjiang and Liaoning samples have 31 identical VOCs (Figure 2B). Detected VOCs were classified into 10 chemical classes, including alkane hydrocarbons, aromatic hydrocarbons, other volatile hydrocarbons, aldehydes, aromatic aldehyde, alcohols, organic acids, fatty acids, phenol-containing, and others. The qualitative analysis was performed by class percentage, as shown in Figure 3A.

Alkanes are the primary class in the rice from three areas. Among them, alkane hydrocarbons were the highest concentration, up to 41.3% in Liaoning rice samples, compared to 27.9% and 18.6% in Heilongjiang and Jilin rice samples. Aldehydes are the second largest group in Heilongjiang and Liaoning rice samples, with 14.2% and 25.3%. At the same time, Jilin rice samples showed the highest composition at 61.5% of aldehydes. Aldehydes are critical substances in the formation of rice flavor and play an important role in rice flavor [19]. Samples from Heilongjiang were found to be enriched in other volatile hydrocarbons (23.0%), aldehydes, aromatic aldehydes (6.7%), and alcohols (9.1%) in comparison with Liaoning and Jilin rice samples. Similarly, phenol-containing (1.5%) was more abundant in samples from Heilongjiang than in the Liaoning and Jilin rice samples. At the same time, aromatic hydrocarbons were found in more significant amounts in Liaoning (2.0%) and Heilongjiang (1.8%) than in Jilin (0.7%). The relative levels of organic acids and fatty acids were the same in all rice samples.

The unsupervised hierarchical clustering and K-means clustering quantitative analysis of 10 chemical classes in rice samples from Heilongjiang, Jilin, and Liaoning provinces are shown in Figure 3B. Heilongjiang and Jilin rice samples clustered together and away from Liaoning rice samples. The Liaoning rice samples showed a higher quantity of other volatile hydrocarbons, alkane hydrocarbons, aromatic hydrocarbons, alcohols, aromatic aldehyde, and fatty acids. Higher amounts of phenol-containing and others were found in the Heilongjiang rice samples. Higher quantities of aldehydes and organic acids were present in the Jilin rice sample.

The unsupervised hierarchical clustering and K-means clustering quantitative analysis of 10 chemical classes in rice samples from Heilongjiang, Jilin, and Liaoning provinces are shown in Figure 3B. Heilongjiang and Jilin rice samples clustered together and away from Liaoning rice samples. The Liaoning rice samples showed a higher quantity of other volatile hydrocarbons, alkane hydrocarbons, aromatic hydrocarbons, alcohols, aromatic aldehyde, and fatty acids. Higher amounts of phenol-containing and others were found in the Heilongjiang rice samples. Higher quantities of aldehydes and organic acids were present in the Jilin rice sample.

The unsupervised hierarchical clustering and K-means clustering quantitative analysis of 10 chemical classes in rice samples from Heilongjiang, Jilin, and Liaoning provinces are shown in Figure 3B. Heilongjiang and Jilin rice samples clustered together and away from Liaoning rice samples. The Liaoning rice samples showed a higher quantity of other volatile hydrocarbons, alkane hydrocarbons, aromatic hydrocarbons, alcohols, aromatic aldehyde, and fatty acids. Higher amounts of phenol-containing and others were found in the Heilongjiang rice samples. Higher quantities of aldehydes and organic acids were present in the Jilin rice sample.

### 3.3. Partial Least Square Discriminant Analysis

The unsupervised PCA analysis of VOCs obtained from all rice samples is shown in Figure 4A. The two principal components accounted for 95.2% of the total variance (Figure 4A). Pearson correlation analysis is performed on the sample groups to correlate between the sample groups. There is a strong positive correlation between each sample group and the corresponding sample group (Figure 4B).

PLS-DA is a supervised pattern identification approach that searches for relationships and suitable classes for each sample [23]. Thus, PLS-DA continued to predict to the class members and explained the majority of the modifiability of the VOCs of Heilongjiang, Jilin, and Liaoning rice samples (Appendix A). The Heilongjiang, Jilin, and Liaoning rice samples identified a clear separation. As presented in Appendix A, three separate clusters were found in the PLS-DA score plot, wherein the two components together accounted for 95.2% of the total variation. The first component explained (76.6%) the variation between the Heilongjiang, Jilin province rice samples from the Liaoning rice samples. The second part explains the difference (18.6%) among Heilongjiang and Jilin rice samples. The corresponding loadings plot accounted for the discovered division among the rice samples from three regions (Appendix A). Validation of the prediction accuracy and fit of the polynomial model was performed using five-fold cross-validation (Appendix A). The results show cumulative values of PLS-DA with R^2^ = 0.992, Q^2^ = 0.983, and accuracy = 1.0, presenting an excellent fitting model. Further validation of the supervised model was performed with 1000 permutation trials (Appendix A). These distribution results indicated that the ability of the best model to predict the VOCs in the sample group was found to be significant at *p* < 0.001. The established model was applied to predict rice samples from Heilongjiang, Jilin, and Liaoning. As a result, all trial samples were accurately appointed with 100% correctness (Figure 5A–F). In addition, the area under the curve (AUC) is 1, meaning that the receiver operating characteristic (ROC) analysis on the basis of PLS-DA is reasonable, with great susceptivity (100%) and characteristics (Figure 5A–F).

### 3.4. Specific VOC Compounds Analysis

Orthogonal projection latent structure discriminant analysis (OPLS-DA) is an essential method for dimensionality reduction and identification of spectral features that drive group separation. It is a supervised modeling method used to discriminate among the varieties in a dataset relevant to predicting group labels and variations unrelated to predicting group labels [25]. Figure 6A shows the OPLS-DA biplot for the discrimination between Heilongjiang, Jilin, and Liaoning rice samples. In total, 33 VOCs were used as the independent variable X for OPLS-DA analysis, X variance R^2^_X_ = 0.986, Y variance R^2^_Y_ = 0.998, and cross-validation predictive power Q^2^ = 0.964, R^2^_X_ − R^2^_Y_ < 0.3, indicating that the model is reliable. Thus, the variable importance for the projection (VIP) applied to determine specific VOCs for geographical discrimination of rice samples [15] and obtained from the corresponding loadings plot is shown in Figure 6B, Table 1. In total, 14 VOCs with VIP > 1 were selected as marked compounds to identify rice samples from three regions.

The numbers corresponding to the volatile compounds’ names are as follows:

1. Decane; 2. Undecane; 3. 2-methyl-decane; 4. Dodecane; 5. Tridecane; 6. Tetradecane; 7. Hexadecane; 8. 2,6,10,14-tetramethyl-hexadecane; 9. Styrene; 10. 2,6-dimethyl-decane; 11. 2-methyl-dodecane; 12. 2,6,10-trimethyl-dodecane; 13. Heptanal; 14. (*E*,*E*)-2,4-nonadienal; 15. Nonanal; 16. (*E*)-2-heptenal; 17. Decanal; 18. Pentadecanal; 19. Hexadecanal; 20. (*E*)-6,10-dimethyl-5,9 undecadien-2-one; 21. 6,10-dimethyl-2-undecanone; 22. Hexanol; 23. 1-octanol; 24. n-heptadecanol; 25. 2-hexyl-1-decanol; 26. 3,7,11-trimethyl-1-dodecanol; 27. Octanoic acid; 28. n-Hexadecanoic acid; 29. Hexanoic acid; 30. Nonanoic acid; 31. Benzyl alcohol; 32. Toluene; 33. 2-butyl-1-octanol.

Generally, alkane and alkenes compounds have no contribution to the rice aroma or flavor; these compounds were considered to be related to lipid decomposition [4]. In this study, eight alkane compounds (tridecane, tetradecane, hexadecane, 2,6,10,14-tetramethyl-hexadecane, styrene, 2,6-dimethyl-decane, 2-methyl-dodecane, and 2,6,10-trimethyl-dodecane) were detected as discriminative markers to identify Heilongjiang, Jilin, and Liaoning rice samples (Figure 7, Table 1). Except for 2,6,10,14-tetramethyl-hexadecane, the other seven compounds were higher in Liaoning rice samples than in the Heilongjiang and Jilin rice samples. The average concentration of tridecane, tetradecane, hexadecane, styrene, 2,6-dimethyl-decane, 2-methyl-dodecane, and 2,6,10-trimethyl-dodecane) in Liaoning rice samples were 56.73, 456.68, 14.96, 28.36, 12.15, 178.96, and 149.58 μg/kg, respectively. Dong et al. [4] and Ch et al. [1] also demonstrated the occurrence of alkanes in Chinese rice. Alkanes are also highly sensitive to environmental factors such as light conditions, temperature, and humidity [4]. Thus, the concentration of alkanes in rice varies from region to region. In conclusion, although alkanes did not contribute to rice flavor, they played an important role in rice VOC identification.

Aldehydes are the key volatile odorant in the formation of rice flavor, which plays an essential role in rice flavor [26]. Relevant studies show that the odor activity value of aldehydes accounts for 97% of the odor activity value of various flavor substances in rice [27]. In the present study, (*E*)-2-heptenal and decanal were found as a marked odorant with the highest concentration in Liaoning rice samples compared to the Heilongjiang and Jilin. The average concentration of (*E*)-2-heptenal and decanal was 13.09 and 171.43 μg/kg in Liaoning rice samples. (*E*)-2-heptenal is a significant VOC in rice [28], it is produced by the decomposition of hydrogen peroxide [28] and contributes to rice’s herbaceous flavor character [29,30]. The decanal contributed to rice as a fatty and citrusy flavor [28].

On the other hand, the alcohols had higher odor threshold odorants, given rice banana sweetness and slight alcohol flavor [28,31]. The hexanol,1-octanol and 2-hexyl-1-decanol were found as odorants with the highest concentration in Liaoning rice samples compared to the Heilongjiang and Jilin. The average concentration of these compounds was 40.53, 10.82, and 31.15 μg/kg in Liaoning rice samples. However, the nonanoic acid was a marked odorant with the highest concentration in Jilin rice samples compared to Heilongjiang and Liaoning. The average concentration of these was 2.75 μg/kg. In this study, higher concentrations of nonanoic acid were present in the Jilin rice sample. The average concentration of nonanoic acid in the Jilin rice sample was 1.7 μg/kg, and it contributed to rice as a waxy type of odor [28].

## 4. Conclusions

This research analyzed the geographical discrimination of 30 rice samples from Northeast China (Heilongjiang, Jilin, and Liaoning province) by HS-SPME-GC-MS and multivariate statistics. The optimal conditions for HS-SPME of VOCs in rice are as follows: rice weight is 3.0 g, the extraction temperature is 30 °C, and the extraction time is 30 min. Overall, 33 VOCs were detected. PCA analysis of VOCs showed that the rice samples from Heilongjiang, Jilin, and Liaoning were significantly distinguished on the score map. A PLS-DA model was constructed, in which the first two principal components explained 95.2% of the total variance, and all samples were correctly classified. A total of 14 characteristic compounds were screened by OPLS-DA analysis for differentiation in rice from Heilongjiang, Jilin, and Liaoning provinces. Applying nontargeted VOC metabolomics to determine rice from different geographic regions has proven to be a hopeful and mighty method for confirming the assuredness and origin of rice from Heilongjiang, Jilin, and Liaoning provinces. The potential for establishing sample banks from other regions is obvious and may result in changes in the authenticity of rice in the future. In addition, this method provides an effective way for the origin identification of other foods (fruits, vegetables, dairy, etc.). Therefore, it can effectively avoid fake and shoddy favorite and high-quality agricultural products and geographical indication of agricultural products.

## Figures and Tables

**Figure 1 foods-11-01695-f001:**
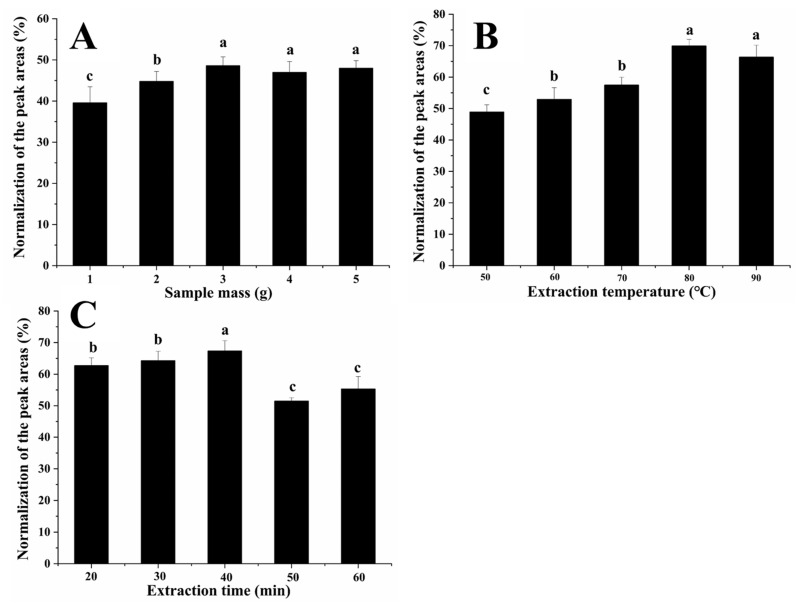
Effect of (**A**) sample weight, (**B**) extraction temperature, and (**C**) extraction time on the extraction efficiency. Different small letters a, b, c in the figure indicate the significant difference between different treatment groups (*p* < 0.05).

**Figure 2 foods-11-01695-f002:**
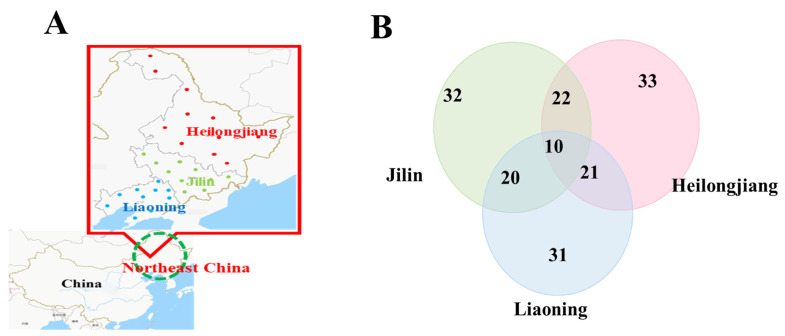
(**A**) Differentiation of Heilongjiang, Jilin, and Liaoning rice. (**B**) Sample gathering provinces from Heilongjiang, Jilin, and Liaoning. The number in the figure represents the number of VOCs detected in rice in this region, and the numbers with overlapping circles represent the number of VOCs common to rice in these regions.

**Figure 3 foods-11-01695-f003:**
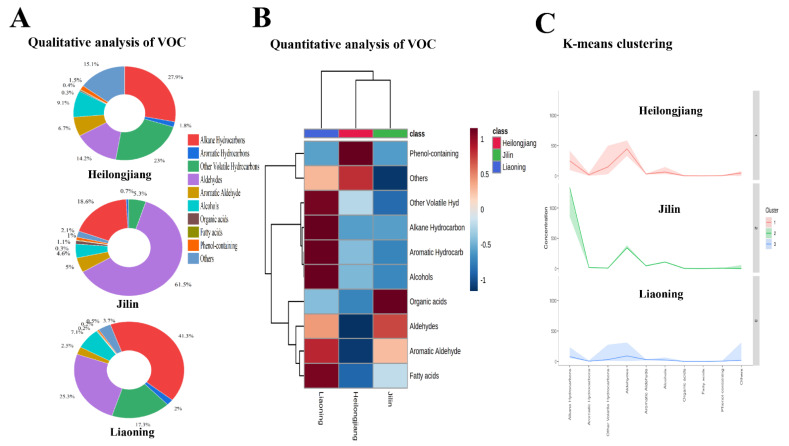
(**A**) Qualitative analysis of VOCs, alkane hydrocarbons, aromatic hydrocarbons, other volatile hydrocarbons, aldehydes, aromatic aldehydes, alcohols, organic acids, fatty acids, phenol-containing, other compounds in a % pie chart. (**B**) Quantitative analysis of VOCs compounds analyzed by HCA. (**C**) K-means clustering of VOC compounds.

**Figure 4 foods-11-01695-f004:**
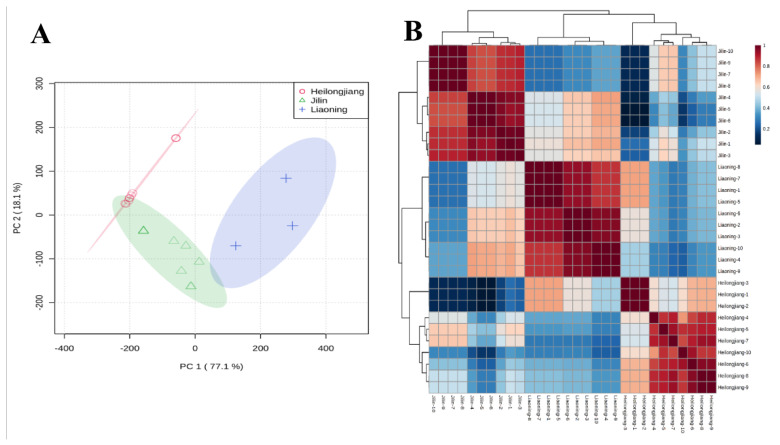
(**A**) PCA analysis for Heilongjiang (red color), Jilin (green color), and Liaoning (blue color) rice samples. (**B**) Pearson correlation of rice samples from Heilongjiang, Jilin, and Liaoning.

**Figure 5 foods-11-01695-f005:**
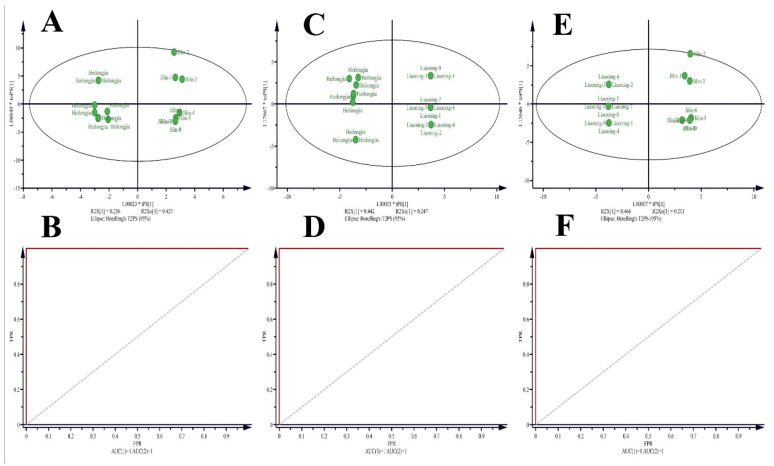
Predictive class probabilities for samples and ROC plots based on the cross-validation (CV) performance. (**A**,**B**) Heilongjiang and rice samples, AUC = 1. (**C**,**D**) Heilongjiang and Liaoning rice samples, AUC = 1. (**E**,**F**) Jilin and Liaoning rice samples, AUC = 1.

**Figure 6 foods-11-01695-f006:**
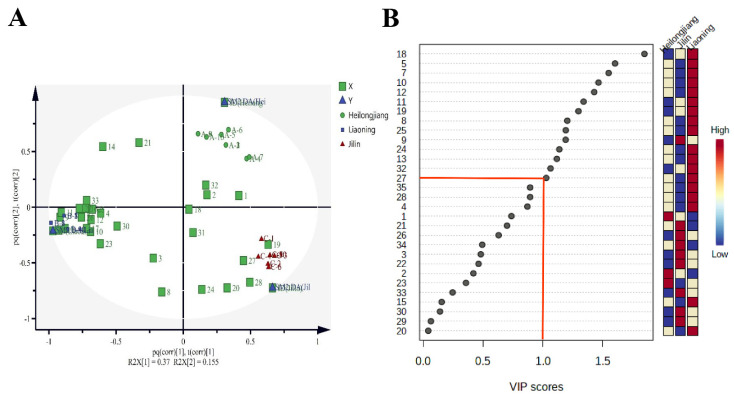
(**A**) Biplot for the discrimination between Heilongjiang (yellow color), Jilin (pink color), and Liaoning (origin color) rice samples. (**B**) Important features identified by OPLS-DA. The colored boxes on the right indicate the relative concentrations of the corresponding metabolite in each group under study. The red line in Figure B indicated that VIP > 1 by OPLS-DA.

**Figure 7 foods-11-01695-f007:**
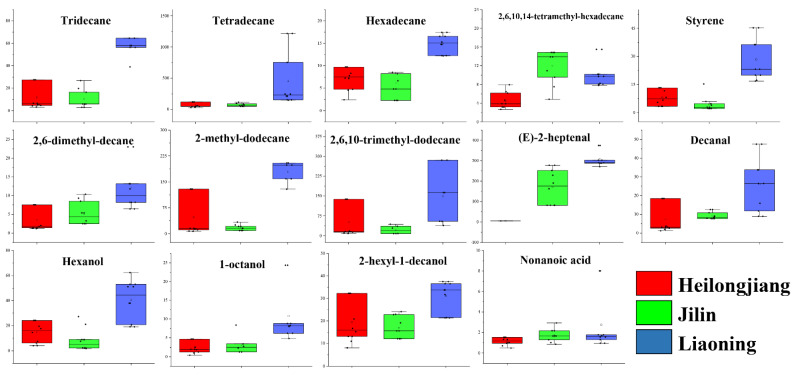
Box plot for distinguishing VOCs in Heilongjiang, Jilin, and Liaoning rice.

**Table 1 foods-11-01695-t001:** Distinguishing VOCs in Heilongjiang, Jilin, and Liaoning.

No.	Compound	CAS	Formulate	VIP	*p*-Value	FDR
1	Tridecane	629-50-5	C_13_H_28_	1.6043	5.407 × 10^−14^	1.622 × 10^−12^
2	Tetradecane	629-59-4	C_14_H_30_	1.5507	2.841 × 10^−10^	1.704 × 10^−9^
3	Hexadecane	544-76-3	C_16_H_34_	1.2049	1.505 × 10^−5^	3.763 × 10^−5^
4	2,6,10,14-tetramethyl-hexadecane	638-36-8	C_20_H_42_	1.1911	6.349 × 10^−6^	1.807 × 10^−5^
5	Styrene	100-42-5	C_8_H_8_	1.4660	1.709 × 10^−11^	1.709 × 10^−10^
6	2,6-dimethyl-decane	13150-81-7	C_12_H_26_	1.3412	3.378 × 10^−5^	6.334 × 10^−5^
7	2-methyl-dodecane	1560-97-0	C_13_H_28_	1.4299	2.885 × 10^−11^	2.163 × 10^−10^
8	2,6,10-trimethyl-dodecane	3891-98-3	C_15_H_32_	1.1175	2.613 × 10^−5^	6.029 × 10^−5^
9	(*E*)-2-heptenal	18829-55-5	C_7_H_12_O	1.8503	1.473 × 10^−12^	2.209 × 10^−11^
10	Decanal	112-31-2	C_10_H_20_O	1.2985	3.33 × 10^−5^	6.334 × 10^−5^
11	Hexanol	626-93-7	C_6_H_14_O	1.1383	5.386 × 10^−6^	1.795 × 10^−5^
12	1-octanol	111-87-5	C_8_H_18_O	1.1921	0.0001302	0.0002297
13	2-hexyl-1-decanol	2425-77-6	C_16_H_34_O	1.0285	0.0003811	0.0006351
14	Nonanoic acid	112-05-0	C_9_H_18_O_2_	1.0650	0.0087844	0.011458

## Data Availability

All data included in this study are available upon request by contact with the corresponding author.

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
