# Peer review of "Fingerprinting of Volatile Organic Compounds for the Geographical Discrimination of Rice Samples from Northeast China"

_foods, 2022, doi:10.3390/foods11121695_

Round 1

Reviewer 1 Report

Fingerprinting of volatile organic compounds for the geographical discrimination of rice samples from Northeast China

In the submitted paper, a non-targeted volatile compound (VOCs) metabolomics method was applied to research the phenotype of rice from Heilongjiang, Jilin, and Liaoning provinces using headspace gas chromatography mass spectrometry HS-SPME/GC-MS with multivariate pattern recognition analysis (unsupervised HCA, PCA and supervised PLS-DA), thereby identifying the potential variation markers of VOCs in rice.

This is an extensive research, with a lot of numerical analysis based on HS-GC–MS data.

Thematically the work is interesting for the researchers and professionals and the proposed  manuscript is relevant to the scope of the journal.

I found it appropriate for publication in the Foods journal, but only after some modifications and clarification from the Authors.

Abstract, title and references          

Abstract is concise and clearly written, with a good command of English, and clear representation of the aim of the paper. Containing exactly 200 words, it can be said that it does meet the demands of the journal Foods (200 max). Furthermore, it is adequately structured: background of the proposed research, the method used, and main conclusions were mentioned.

The title of the paper adequately reflects the subject under investigation in the proposed study.

The literature review is comprehensive and properly done. References are numbered in order of appearance in the text, as demanded by formatting rules of the journal. Although there is no limitation in the number of references, a reference list of 32 citations is completely sufficient to cover the topic proposed.

Introduction  

The authors clearly represented the importance of the issue described. In the final paragraphs of the introductory section the authors explain what the core of their research is. The novelty of the work must be more clearly demonstrated.

Materials and Methods        

The authors adequately described the sampling procedure and sample preparation, extraction of volatiles, optimization of the HS-SPME method, GC-MS parametes used, and mutlivariate data treatment.

Results and Discussion         

The meaning, relevance and importance of the obtained results are explained in the same section. Furthermore, they were adequately compared with previous studies.

The significance of the Work: Given the large number of analyzed data, this is an interesting study with a possible significant impact in this area.

Statistical interpretation of the analytical data must be more properly presented.

The verification of the model should be performed. Model validation is possibly the most important step in the model building sequence. Unfortunately, a high R2 value does not guarantee that the model fits the data well. The residuals from a fitted model are the differences between the responses observed and the corresponding prediction of the response computed using the regression function. If the model fit to the data were correct, the residuals would approximate the random errors that make the relationship between the explanatory variables and the response variable a statistical relationship.

Conclusions   

Conclusions are clear and strongly supported by the results obtained.

This is an extensive research, with a lot of numerical analysis based on HS-GC–MS data.

Other Specific Comments: The work is properly presented in terms of the language. The work presented here is very interesting and well done; it is presented in a compact manner.
The results are presented in a logical sequence and the discussion and analysis of the results are properly elaborated.
The manuscript should be improved from technical/graphical viewpoint, as described in the pdf file.

Author Response

Response: Thank you very much for your positive comments on our manuscript. According to your comments, the manuscript has been modified. Please check the revised manuscript.

Reviewer 2 Report

I enjoyed reading the manuscript entitled “Fingerprinting of volatile organic compounds for the geographical discrimination of rice samples from Northeast China”. Manuscript is very well organized but some proposed changes should be done. In my point of view, is minor comments.

My comments are as follow:

  • In line 163, I tried to enter to http://www.metaboanalyst.cn/ but was no possible.
  • At the end of the conclusion section, you stated: “The potential for establishing sample banks from other regions is obvious and may lead to changes in the authenticity of rice in the future.”. I suggest to include a sentence to stablish pros and cons related to applicate this methodology to another foods (fruits, vegetables, dairy, etc.).

.

Author Response

I enjoyed reading the manuscript entitled "Fingerprinting of volatile organic compounds for the geographical discrimination of rice samples from Northeast China." Manuscript is very well organized but some proposed changes should be done. In my point of view, is minor comments. My comments are as follows:

  1. Comment: English language and style are fine/minor spell check required.

Response: Thank you for your suggestions. Our native English speak colleague Amjad Sohail helped revise the English grammar in the manuscript. In the acknowledged part of the manuscript, we sincerely thanks him for his help. Please refer to the revised manuscript. Please refer to line 82.

  1. Comment: In line 163, I tried to enter to http://www.metaboanalyst.cn/ but was no possible.

Response: Thank you for your suggestions. We apologize for a small error in the URL name and the correct web address, such as "https://www.metaboanalyst.ca/."The web page opened by this URL is shown in Figure S1. We provided the correct URL in the revised manuscript. Please refer to line 162.

Figure S1. Web of metaboanalyst

  1. Comment: At the end of the conclusion section, you stated: “The potential for establishing sample banks from other regions is obvious and may lead to changes in the authenticity of rice in the future.”. I suggest to include a sentence to stablish pros and cons related to applicate this methodology to another foods.

Response: Thank you for your suggestions. We add the following content to the conclusion of the manuscript.“In addition, this method provides an effective way for the origin identification of other foods  (fruits, vegetables, dairy, etc.). Therefore, it can effectively avoid fake and shoddy favorite and high-quality agricultural products and geographical indication of agricultural products.” Please refer to lines 366-369.

Reviewer 3 Report

The manuscript is incomplete, several tables and figures are missing:
Line: 101: Table S1 is not available
Line: 214: Figure S1 is not available
Line: 215: Table S2 is not available
Lines: 274-285: Figure S2 is not available
Which makes it difficult to understand the reason for undertaking such research.
Please precisely define the purposefulness of the research undertaken. Despite numerous reports, it may not seem necessary to geographically identify the rice's origin if it does not entail health risks for consumers.  Knowledge gap as well as the work novelty must be highlighted before listing your objectives.
I cannot recommend this manuscript for printing in this version, although the chemical methods are clearly described and properly carried out. 

Author Response

Response to Reviewer 3 Comments

  1. Comment: The manuscript is incomplete, several tables and figures are missing:
    Line: 101: Table S1 is not available
    Line: 214: Figure S1 is not available
    Line: 215: Table S2 is not available
    Lines: 274-285: Figure S2 is not available

Which makes it difficult to understand the reason for undertaking such research.

Response: Thank you for your suggestions. Table S1, Table S2, Figure S1, and Figure S2 were provided as SUPPLEMENTARY INFORMATION in this manuscript. We submitted SUPPLEMENTARY INFORMATION separately. We sincerely apologize for the SUPPLEMENTARY INFORMATION that was missing. Table S1, Table S2, Figure S1, and Figure S2 are as follows:

Table S1 Geographical origin of the rice samples

Samples

Variety

Region

Prefecture

1

Longjing 31

Heilongjiang

Qiqihar

2

Longjing 39

Heilongjiang

Qiqihar

3

Longjing 46

Heilongjiang

Qiqihar

4

Longjing 71

Heilongjiang

Qiqihar

5

Suijing 306

Heilongjiang

Suihua

6

Longke 3

Heilongjiang

Qiqihar

7

Jinyu 1

Heilongjiang

Qiqihar

8

Suijing 18

Heilongjiang

Suihua

9

Suijing 28

Heilongjiang

Suihua

10

Suijing 302

Heilongjiang

Suihua

11

Jijing303

Jilin

Changchun

12

Jijing525

Jilin

Changchun

13

Jijing816

Jilin

Changchun

14

Jijing 830

Jilin

Tonghua

15

Jiyang 1

Jilin

Tonghua

6

Jiyang 100

Jilin

Tonghua

17

Tonghe 877

Jilin

Tonghua

18

Tonghe 822

Jilin

Tonghua

19

Tongyuan 568

Jilin

Tonghua

20

Tongke 29

Jilin

Tonghua

21

Liaojing 212

Liaoning

Shenyang

22

Liaojing 436

Liaoning

Shenyang

23

Liaojing 433

Liaoning

Shenyang

24

Liaojing 419

Liaoning

Shenyang

25

Liaoxing 21

Liaoning

Shenyang

26

Yanjing 927

Liaoning

Panjin

27

Yanjing 337

Liaoning

Panjin

28

Yanfeng 47

Liaoning

Panjin

29

Yanjing 939

Liaoning

Panjin

30

Yanjing 219

Liaoning

Panjin

Table S2 VOCs identified in the rice sample with GC–MS. (μg/kg)

NO

Compound

CAS

Molecular Formula

RI

Heilongjiang

Jilin

Liaoning

1

Decane

 124-18-5

C10H22

1000

0.58−3.24

0.35−5.42

0.28−0.62

2

Undecane

1120-21-4

 C11H24

1100

1.78−6.59

0.71−12.97

2.29−3.43

3

2-methyl-decane

6975-98-0

C11H24

1064

0.46−7.14

2.1−32.98

15.02−15.21

4

Dodecane

112-40-3

C12H26

1200

8.87−22.41

4.72−28.37

17.45−59.48

5

Tridecane

629-50-5

C13H28

1300

3.15−27.48

2.98−26.78

56.34−64.59

6

Tetradecane

629-59-4

C14H30

1400

28.91−119.44

49.2−115.99

153.71−246.26

7

Hexadecane

 544-76-3

C16H34

1600

2.44−9.72

2.28−19.27

12.26−16.56

8

2,6,10,14-tetramethyl-hexadecane

 638-36-8

C20H42

1792

2.69−7.93

4.85−14.85

8.06−15.51

9

Styrene

100-42-5

C8H8

1254

3.27−13.22

2.13−15.29

23.17−45.29

10

2,6-dimethyl-decane

13150-81-7

C12H26

1112

1.15−7.54

2.49−10.35

6.43−23.0

11

2-methyl-dodecane

 1560-97-0

C13H28

1264

6.89−129.06

9.33−33.9

158.78−204.51

12

2,6,10-trimethyl-dodecane

3891-98-3

C15H32

1366

8.72−138.3

8.13−43.86

54.8−286.06

13

Heptanal

111-71-7

 C7H14O

1182

8.70−8.70

4.70−9.75

n.d.

14

(E,E)-2,4-nonadienal

5910-87-2

C9H14O

1216

9.71−71.91

0.58−12.93

28.31−61.67

15

Nonanal

124-19-6

C9H18O

1104

8.29−17.85

2.51−44.89

n.d.

16

(E)-2-heptenal

 18829-55-5

C7H12O

1334

5.03−5.03

81.48−277.59

289.36−374.42

17

Decanal

112-31-2

C10H20O

1206

1.18−18.40

7.69−12.53

8.94−47.46

18

Pentadecanal

316249

C15H30O

1715

1.59−6.82

0.46−18.36

2.36−2.88

19

Hexadecanal

629-80-1

C16H32O

1817

0.52−2.10

1.44−2.84

0.81−0.88

20

(E) 6, I0-dimethyl-5,9 undecadien-2-one

3796-70-1

C13H22O

1876

4.79−15.45

12.62−29.42

11.50−19.01

21

6,10-dimethyl-2-undecanone

1604-34-8

C13H26O

1408

13.76−25.85

6.99−20.19

12.11−19.78

22

Hexanol

626-93-7

C6H14O

1211

3.90−24.12

1.71−27.2

19.03−51.06

23

1-octanol

111-87-5

C8H18O

1558

0.42−4.66

1.28−8.39

6.22−24.36

24

n-heptadecanol

1454-85-9

C17H36O

1984

0.27−0.85

1.26−3.46

1.49−1.69

25

2-hexyl-1-decanol

2425-77-6

C16H34O

1504

8.10−32.28

12.09−24.13

21.35−37.47

26

3,7,11-trimethyl-1-dodecanol

6750-34-1

C15H32O

1571

1.88 −5.75

1.13−7.99

12.54−49.56

27

Octanoic acid

124-07-2

C8H16O2

2050

0.14 −0.40

0.27−5.78

0.26−0.31

28

n-Hexadecanoic acid

57-10-3

C16H32O2

1968

1.05 −2.03

2.02−8.28

1.58−2.38

29

Hexanoic acid

142-62-1

C6H12O2

1849

0.09 −0.09

n.d.

0.42−0.42

30

Nonanoic acid

112-05-0

C9H18O2

2173

0.48 −1.52

0.85−2.92

0.95−8.01

31

Benzyl alcohol

100-51-6

C7H8O

1877

1.89 − 6.22

3.61−5.57

3.85−5.21

32

Toluene

108-88-3

C7H8

1036

0.88 −11.29

0.61−3.05

1.17−1.96

33

2-butyl-1-octanol

735273

C12H26O

1277

11.95−80.42

6.69−41.94

55.07−79.89

RI: retention index.

n.d.: not detected.

Figure S1. Total ion chromatogram of VOCs profile of Heilongjiang, Liaoning, and

Jilin rice samples were obtained from HS-SPME-GC-MS.

Figure S2. Significant differences in VOCs profile of Heilongjiang, Jilin, and Liaoning rice samples A) PLS-DA scores plot for the first component (76.6%) Vs second component (18.6%), indicating discrimination between Heilongjiang (red color), Jilin (green color), and Liaoning (blue color) rice samples. B) Three-dimensional PLS-DA scores plot. C) Three-dimensional PLS-DA loading plot. D) PLS-DA classification using the different number of components. Bar plots show the three performance measures (prediction accuracy, R2, and Q2) using different components. The red '*' indicates the best values of the currently selected measures (Q2). E) PLS-DA model validation by permutation tests based on the separation distance. The p-value based on permutation is P< 0.001 (0/1000).

  1. Comment: Please precisely define the purposefulness of the research undertaken. Despite numerous reports, it may not seem necessary to geographically identify the Rice's origin if it does not entail health risks for consumers. Knowledge gap as well as the work novelty must be highlighted before listing your objectives.

Response: Thank you for your suggestions. As described in the preface to the manuscript, Rice with a geographic origin label in China, such as Wuchang Rice in Heilongjiang, Panjin rice in Liaoning, and Baijinxiang rice in Jilin, are considered high-quality Rice and favored by consumers [1]. While there are significant differences in rice quality and composition across regions, it seems impossible to discriminate Rice's geographic origin by appearance or regular analysis [2]. In addition, volatile organic compounds (VOCs) are the diagnostic aspects of rice quality that can determine consumers' acceptance. They are also essential indicators of geographical origins. Therefore, in the current study, a non-targeted VOCs metabolomics method was applied to research the phenotype of Rice from Heilongjiang, Jilin, and Liaoning provinces. Specifically, HS-SPME and GC-MS and multivariate pattern recognition analysis identified the potential variation markers of VOCs in rice.

  1. Dong KL.; Mo C.; Lee DK.; Long NP.Kwon SW. Non-Destructive Profiling of Volatile Organic Compounds Using Hs-Spme/Gc–Ms and its Application for the Geographical Discrimination of White Rice. J Food Drug Anal. 2018, 26,260.DOI:10.1016/j.jfda.2017.04.005
  2. Suzuki Y.; Chikaraishi Y.; Ogawa NO.; Ohkouchi N.Korenaga T. Geographical Origin of Polished Rice Based On Multiple Element and Stable Isotope Analyses. Food Chem. 2008, 109,470-475.DOI:10.1016/j.foodchem.2007.12.063

  3. Comment: I cannot recommend this manuscript for printing in this version, although the chemical methods are clearly described and properly carried out. 

Response: First of all, thank you very much for your affirmation of the writing of this manuscript. Since the absence of the attached materials SUPPLEMENTARY INFORMATION makes you confused about the article, we have supplemented the SUPPLEMENTARY INFORMATION and further modified and adjusted the grammar and writing standards of the article in detail.
